# Critical evaluation of kinetic schemes for coagulation

**Alexandre Ranc[1], Salome Bru[2], Simon Mendez[3], Muriel Giansily-Blaizot[1], Franck Nicoud[3], Rodrigo Méndez Rojano** [3]*

1 Department of Haematology Biology, CHU, Univ Montpellier, Montpellier, France, 2 Polytech, Univ Montpellier, Montpellier, France, 3 IMAG, Univ Montpellier, CNRS, Montpellier, France

* fsae.unam.rodrigo@gmail.com

**Data Availability Statement:** All experimental and simulation data are available at Github Repository at https://github.com/rodrigomrxvi/CriticalEvaluationCoagulation.

## Abstract

Two well-established numerical representations of the coagulation cascade either initiated by the intrinsic system (Chatterjee et al., PLOS Computational Biology 2010) or the extrinsic system (Butenas et al., Journal of Biological Chemistry, 2004) were compared with thrombin generation assays under realistic pathological conditions. Biochemical modifications such as the omission of reactions not relevant to the case studied, the modification of reactions related to factor XI activation and auto-activation, the adaptation of initial conditions to the thrombin assay system, and the adjustment of some of the model parameters were necessary to align in vitro and in silico data. The modified models are able to reproduce thrombin generation for a range of factor XII, XI, and VIII deficiencies, with the coagulation cascade initiated either extrinsically or intrinsically. The results emphasize that when existing models are extrapolated to experimental parameters for which they have not been calibrated, careful adjustments are required.

## 1 Introduction

In recent decades, numerical modeling of the coagulation cascade has been used to understand the complex dynamics of thrombin formation [1–3], to identify coagulation factor interactions [4], and to investigate the dynamics of thrombus formation in pathological conditions [5, 6].

Such numerical models of the coagulation cascade are usually used to shed light on the mechanisms leading to clotting disorders. Most of these models aim to reproduce the thrombin generation process, which results from the balance between prothrombin conversion and thrombin inactivation [7]. Thrombin generation can be summarized *in vivo* with the following enzymatic steps based on [8]. Tissue factor (TF) is released by damaged vessels or prothrombotic cells including tumor cells, activated mono-nuclear cells, or micro-vesicles. TF forms a complex with activated factor (F) VII (FVIIa) and activates FX either directly or indirectly through FIX. Then, FXa activates the prothrombin into the key thrombin factor. Prothrombin conversion is self-regulated. Thrombin activates FXI and both the co-factors VIII and V of factors IX and X, respectively, creating feedback loops that increase its generation leading to a thrombin burst. In parallel, thrombin cleaves the fibrinogen into fibrin monomers forming an unstable network which is subsequently stabilized by activated FXIII. On the other hand, the

**Funding:** RMR was supported by Consejo Nacional de Ciencia y Tecnología (CONACyT) (https://conacyt.mx/), Mexico scholarship and the LabEx Numev (convention ANR-10-LABX-20). FN was awarded by the Agence Nationale de la Recherche (ANR) (the French National Research Agency) under the "Investissements d'avenir" programme with the reference ANR-16- IDEX-0006 (https://anr.fr/). The funders had no role in study design, data collection and analysis, decision to publish, or preparation of the manuscript.

**Competing interests:** The authors have declared that no competing interests exist.

coagulation cascade can also be triggered by the contact activation system as a result of the interactions of FXII, kininogen, and prekallikrein with negatively charged surfaces. This coagulation route, also known as the intrinsic pathway, was historically neglected due to the absence of clinical phenotype in cases of FXII deficiency but has now a renewed interest due to promising anti-FXI and anti-FXII drugs to limit thrombosis in situations where blood meets artificial surfaces of medical devices [9].

Recently, numerical models of the coagulation cascade have been used to shed light on mechanisms leading to clotting disorders. For example, Link *et al.* [10, 11] performed a global sensitivity analysis using the coagulation model of Kuharsky and Fogelson [12] that led to the identification of FV as a key modifier of thrombin generation among patients with hemophilia A. Their results were confirmed later with additional experimental assays highlighting the value of using the numerical model to build a synthetic data set in order to test specific hypotheses. Another examples of modeling coagulation disorders are the studies by Brummel-Ziedins *et al.* [13] who developed a model for FVII deficiency, and that of Anand *et al.* [14], in which the authors examined protein C and antithrombin deficiencies by comparing their results to thrombin generation experiments initiated by the TF-VIIa complex [15]. Models have also been used to study new treatment strategies that would be further assessed *in vitro*. For instance, Burghaus *et al.* [16] and Brummel-Ziedins *et al.* [17] investigated the rivaroxaban effects. Adams *et al.* [18] and recently Zavyalova *et al.* [19] worked on the possible outcomes of direct thrombin inhibitors.

Despite great progress in thrombin formation simulations, a direct comparison between numerical models and *in vitro* experiments is frequently inappropriate since the models are constructed for specific and ideal conditions, as pointed out by Link *et al.* [11, 20]. A great example of this issue is the study by Chelle *et al.* [21] in which five coagulation models are used to predict thrombin formation under relevant clinical scenarios. They found that the predictive capabilities of the models were far from acceptable and needed an additional optimization step using genetic algorithms to adapt kinetic parameters and reproduce *in vitro* data. In particular, they suggested that performing patient-specific optimization is needed to obtain accurate model predictions. However, if each new application requires patient-specific calibration, the global functionality of the model is reduced.

A further challenge of numerical models is related to the uncertainty of input parameters (such as reaction rates and initial concentrations). This uncertainty is linked to lack of experimental standardization when characterizing reaction rate values [20]. Therefore, it is not surprising to find reaction rate values spanning several orders of magnitude for the same coagulation reaction [22]. To avoid this issue, some authors have suggested that instead of trying to find a universal set of reactions, reduced coagulation models are enough to reproduce the coagulation dynamics [7, 23, 24]. Nevertheless, a prior calibration step is still needed.

A strategy to increase the robustness of coagulation models can be to assess the sensitivity of thrombin formation to coagulation factor deficiencies [2, 3, 25–27]. By doing such sensitivity studies, the robustness of the reaction scheme can be tested more deeply, and the reaction rate values can be tuned in a more relevant way, making sure that the final model is not only able to represent one regime, but rather a whole variety of situations. In addition, predicting thrombin kinetics under pathological conditions (like factor deficiencies) is the ultimate goal of most mathematical models.

In the present work, we evaluated two well-established mathematical models of the coagulation cascade from Chatterjee *et al.* [2] for the contact pathway and from Hockin *et al.* [1] and Butenas *et al.* [3] for the TF pathway. Three clotting factor deficiencies were studied to investigate if the models could reproduce the dynamics of thrombin generation. For each model, the results were compared with experimental data using calibrated automated thrombography

(CAT) [28]. A range of decreasing concentrations of the coagulation factor FVIII was considered in order to represent the hemophilia A condition. In addition, deficient FXII and FXI plasmas were used to probe the initiation of the coagulation cascade with the intrinsic pathway, which is of special interest to medical devices. To the best of our knowledge, this is the first time that these models are tested for a range of deficient FXII, FXI and FVIII plasmas. The comparison of numerical and experimental data allowed us to quantify the accuracy of the models, identify limitations, and suggest adjustments that improved the predictive capabilities of the kinetic schemes.

Section 2 describes the thrombography assay and computational simulations. Experimental and simulation results are presented in Section 3.1 for a range of concentrations of FXII, FXI, and FVIII triggered through the intrinsic pathway. Section 3.2 shows the results for the range of FVIII concentrations triggered with TF pathway. Finally, the results and the limitations of our work are commented in Section 4.

## 2 Materials and methods

### 2.1 Ethics statement

We obtained approval from the Montpellier University Hospital ethics committee (Comité Local d'Ethique Recherche, agreement number: IRB-MTP_2023_02_202301342). The need for informed consent was waived on the basis that analyses are done on care sample.

### 2.2 Plasma samples

Samples used to characterize the contact pathway were derived from lyophilized plasmas. To avoid inter-individual physiological variations of clotting factor levels, lyophilized Standard Human Plasma (SHP®) [Siemens Healthcare, Erlangen, France] constituted of industrial pooled citrated platelet poor plasmas (PPP) and obtained from at least one hundred healthy donors was used. Lyophilized factor immunodepleted plasmas were industrial pooled citrated PPP with a qualified specific factor activity lower than 1% [Siemens Healthcare, Erlangen, Germany]. According to manufacturer specifications, donors were pretested for PT, APTT, factor II, V, VII, VIII, IX, X, XI, XII levels, which were all within the normal range. SHP was diluted with lyophilized factor immunodepleted plasmas (FXII, FXI, and FVIII), in order to obtain a range of final concentrations from less than 1% to 100% (100%, 50%, 15%, 5%, 1% and < 1% treated as 0% from here on). Silica, mixed with rabbit cephalin in STA®-PTT-A®(Stago, Asnières-sur-Seine, France), was used to initiate the intrinsic pathway.

Inherited deficient frozen plasmas were preferred to lyophilized plasmas to assess the TF coagulation pathway because of higher values for the thrombin generation assay (TGA) parameters. Duchemin *et al*. [29] highlighted the same difference between the lyophilized and immunodepleted haemophiliac plasmas. Frozen plasmas used to assess the TF coagulation pathway were derived from a single patient with hemophilia A (Cryopep, Montpellier, France). To obtain a range of FVIII concentrations from less than 1% to 100% (100%, 50%, 15%, 5%, 1% and < 1%), the hemophilia A plasma was picked with increased amounts of "in-house" fresh frozen PPP plasmas from seven anonymous residual plasmas tested for normal PT, APTT and fibrinogen levels. To avoid any residual platelets, two sequential centrifugations were performed at 2250 g at 20 ˚C for 10 minutes. The "in-house" PPP aliquots were stored at -80 ˚C. The TF coagulation pathway was triggered using the PPP-Reagent LOW (Thrombinoscope®BV, Maastricht, The Netherlands) containing a final TF concentration of 1pM. Low concentrations of TF were chosen to obtain a thrombin generation mainly dependent upon feedback activation loops and FVIII and FIX clotting factors [30].

## 2.3 Calibrated automated measurement of thrombin generation

Thrombin generation was performed in PPP using Fluoroscan Ascent®(FluCa kit, Thrombinoscope®, Synapse BV, Maastrich, The Netherlands), according to the method described by Hemker *et al.* [28] that measures the thrombin generated thanks to a fluorochrome. 80 $\mu$L of plasma were mixed with 20 $\mu$L of STA®-PTT-A®or PPP reagent LOW, according to the coagulation pathway tested, and 20 $\mu$L of fluorescent reagent FluCa kit. The latter contains the calcium chloride which is mandatory to activate the coagulation cascade and the Fluo-Substrate which is a fluorogenic substrate mixed with the FluoBuffer. Fluorescence intensity was detected at wavelengths of 390 nm (excitation filter) and 460 nm (emission filter), every 20 seconds. Each individual sample was analyzed in triplicate simultaneously with a thrombin calibrator as reference for a stable thrombin activity of approximately 600 nM. The calibrator enables the conversion of the fluorescence signal into thrombin concentration. The signal was treated to correct inner filtering effects, such as substrate consumption and abnormal plasma color. Analyses were conducted on Immulon 2HB round-bottom 96-well plates (Stago—Asnières-sur-Seine, France).

The parameters calculated by Thrombinoscope®software included: (i) endogenous thrombin potential (ETP) to evaluate the overall effect on thrombin generation, (ii) the $IIa_{max}$ which corresponds to the largest value of thrombin, (iii) the time to peak ($\tau_{max}$) which is the time required to reach $IIa_{max}$, and (iv) the lag time ($\tau_{lag}$) required for the generation of 10 nM of thrombin [28]. The physical interpretation of these quantities is provided in Fig 1.

## 2.4 Kinetic schemes and mathematical modeling

Two coagulation kinetic models were used in the current study. For the sake of simplicity, the models of Chatterjee *et al.* [2] and Butenas *et al.* [3] are named Int and Ext, respectively, making reference to the intrinsic and extrinsic pathways.

The models were implemented in python, to solve each biochemical scheme, a set of differential equations was obtained by applying the law of mass action to the biochemical reactions belonging to each kinetic model. This representation mimics a closed homogeneous reactor in which the only mechanisms that influence the dynamics of coagulation factors are production and consumption (without diffusive or convective transport of species [31]). The system of equations was then computationally solved to obtain the evolution of coagulation factors in time using the initial conditions displayed in Table 1. Initial conditions were computed from the dilutions used in TGA assays. The nominal initial conditions used in the study were derived accounting for the dilution steps of the calibrated automated thrombinography.

The Int model was published in 2010 and describes the coagulation triggered by the contact pathway [2]. It was built upon the existing model of Hockin *et al.* [1] with additions concerning the contact pathway reactions. The model is triggered by contact activation of factor XII. Since factor immunodepleted plasmas were collected without corn trypsin inhibitor (CTI), the reaction 35 that describes the inhibition of factor XIIa by CTI in Chatterjee *et al.* [2] was suppressed. Moreover, since the thrombin generation is the basis of the analysis performed in the present study, fibrin-related reactions (# 50 to 57 from [2]) were deleted to save computing time without interfering with the thrombin generation. The Ext model was described in the literature by Hockin *et al.* [1] and upgraded in 2004 by Butenas *et al.* [3] with a supplementary reaction that describes the activation of factor X by the activated factor IX (IXa).

The Int and Ext models were modified to align some reactions and kinetic schemes to the studied physiological conditions. The modified formulations were compared with the original numerical models and experimental data from TGA. The rationale for each specific

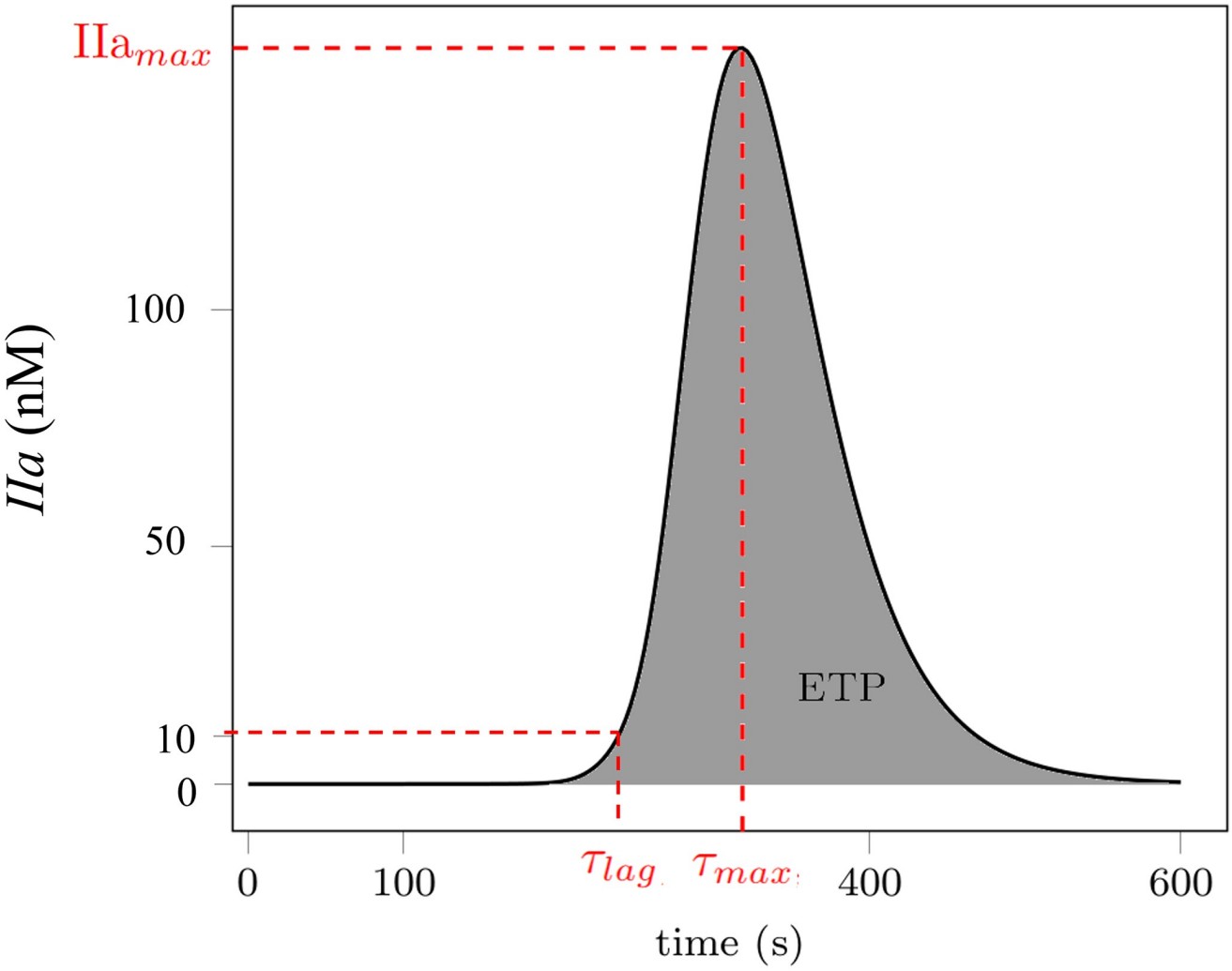

**Fig 1. Variables used for *in vitro* and *in silico* comparison: $IIa_{max}$ (peak thrombin concentration), $\tau_{max}$ (time to peak), $\tau_{lag}$ (time to reach 10 nM of thrombin) and ETP (endogenous thrombin potential) for a thrombin formation curve in time.**

modification is discussed in Section 3. Both the original and modified kinetic schemes used are listed in the S1 Text.

### 2.5 *In vitro* vs *in silico* comparison

To establish a comprehensive comparison between the *in silico* and *in vitro* results, the ETP, $IIa_{max}$ and $\tau_{max}$ variables were considered (see Fig 1). Relative values were computed considering the physiological production of thrombin in PPP as a reference. In the case of Figs 3 and 7, these values are used in order to evaluate the general trend for the different concentrations rather than comparisons using the absolute values.

## 3 Results

### 3.1 Int model for the contact pathway

TGA experiments and simulations using the original Int model were conducted over a range of decreasing concentrations of FXII, FXI, and FVIII. Fig 2 shows the simulation results and

**Table 1. Initial conditions of Int and Ext models, derived from thrombin generation assay dilution.** * TF nominal value, see Section 3.2.

| Factor | Int Model Initial Concentrations [nM] | Ext Model Initial Concentration [nM] |
|---|---|---|
| VII | 6.67 | 6.67 |
| $VII_a$ | 0.0667 | 0.0667 |
| X | 106.67 | 106.67 |
| IX | 60.0 | 60.0 |
| II | 933.0 | 933.0 |
| VIII | 0.4667 | 0.4667 |
| V | 13.33 | 13.33 |
| TFPI | 1.667 | 1.667 |
| AT | 2267.0 | 2267.0 |
| XII | 226.7 | NA |
| PK | 300.0 | NA |
| $C1_{INH}$ | 1667.0 | NA |
| $\alpha_1 AT$ | 30000.0 | NA |
| $\alpha_2 AP$ | 667.0 | NA |
| XI | 20.67 | NA |
| TF* | NA | 0.001 |

experimental TGA data for plasma with 100%, 15%, and 0% for FXII, FXI and FVIII. Four main discrepancies are observed:

- Numerical data at a concentration of 100% show a delay and steeper rise in thrombin formation compared to experimental data.

- When the concentration is less than 100%, the original model cannot capture the trend observed experimentally.

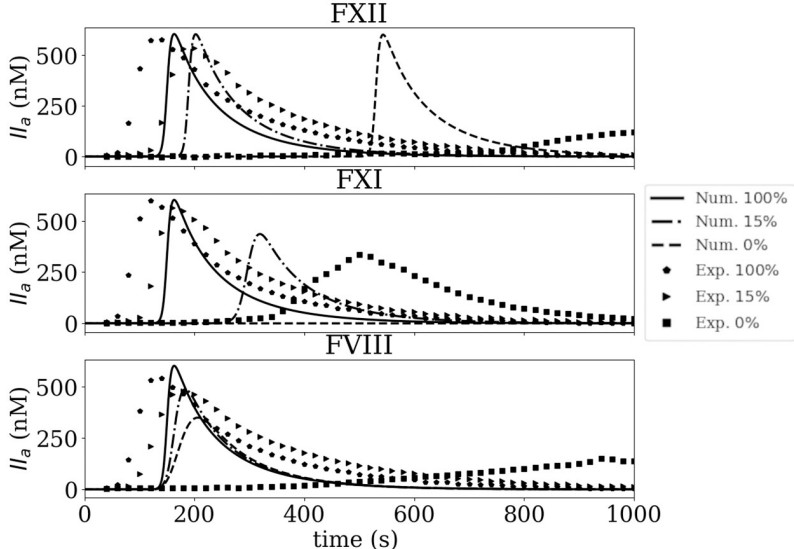

**Fig 2. Experimental thrombin formation (from thrombin generation assay) and simulated thrombin formation using the original Int model for FXII, FXI and FVIII at 100%, 15%, and 0%.**

- In the simulations with 0% of FXII the coagulation cascade should not start at all, but the model predicts thrombin generation.

- For the low concentration of FVIII (0%) simulated thrombin production is significantly larger than experimental data.

Fig 3 shows the evolution of relative ETP, $IIa_{max}$ and $\tau_{max}$ values for all the FXI, FXII, and FVIII concentrations: 0, 1, 5, 15, 50, and 100%. The original and modified numerical model results are shown along with the experimental data. It can be observed that for the FXI case both ETP and $IIa_{max}$ values predicted by the original model show a descending trend steeper than the experimental data. In terms of $\tau_{max}$, the values computed by the original model are significantly larger than the experimental data. Looking at FXII, the $\tau_{max}$ trend is better captured by the original model, however, ETP and $IIa_{max}$ values do not follow the same trend observed experimentally for low concentrations of FXII. In addition, undesired thrombin production is observed for FXII 0% concentration. It is worth saying that *in vitro* thrombin formation for FXII and FXI at 0% cases can be explained by the presence of residual amounts of coagulation factors in the respective immunodepleted plasmas, however, this should not happen numerically. The ETP and $IIa_{max}$ values computed by the original model for FVIII concentrations follow a similar trend as the experimental data, nonetheless, the increased $\tau_{max}$ values observed experimentally for decreasing concentrations of factor VIII are not observed at the same level in the simulation using the original model. The modifications made to improve the model are discussed below.

First, *in silico* thrombin formation for FXII the 0% concentration should not take place since factor FXII activation is the starting point of the coagulation case. The reason for normal, yet delayed, thrombin production at 0% concentration relies on the activation of FX and FIX by activated FVII (FVIIa), independently of TF, by reactions:

$$VIIa + IX \leftrightarrow VIIa = IX \rightarrow VIIa + IXa \qquad (1)$$

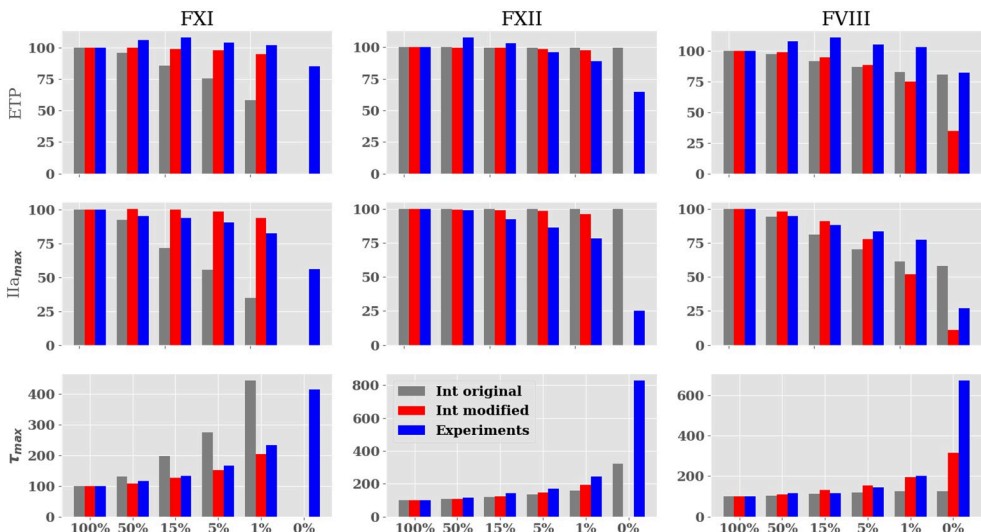

**Fig 3. Evolution of the different parameters (%) for each factor concentration range (%) after contact activation.** GRAY BAR *In silico* Thrombin Generation obtained by Int original; RED BAR obtained by Int modified; BLUE BAR experimental results. In the 0% experimental case for FXII and FXI thrombin is produced since residual amounts of coagulation factors are still present in plasma.

$$VIIa + X \leftrightarrow VIIa = X \rightarrow VIIa + Xa \tag{2}$$

This mechanism was originally proposed by Komiyama *et al.* [32] and incorporated in the Int model. Komiyama *et al.* used 1% of the total FVII concentration in its activated state in order to trigger the reactions and characterize the kinetic rates. The trouble with this rationale is that there are at least three forms of FVII: (i) FVII zymogen, (ii) zymogen-like FVII which corresponds to the 1% circulating activated free form of FVII and (iii) activated FVII which is either bound to TF under physiological conditions or unbound in therapeutic contexts corresponding to the recombinant FVIIa (Novoseven®, Novo Nordisk A/S, Danemark). Reactions (1) and (2) do not reflect physiological mechanisms in the absence of circulating TF, but describe the non-physiological recombinant FVIIa (Novoseven®) interactions [32]. Therefore, both reactions (1) and (2) were deleted in the original and modified Int models. Fig 3 shows that by omitting these reactions, simulated thrombin formation at the 0% FXII case is suppressed.

The activation of FXII due to STA®-PTT-A® reagent was set to $k_1 = 5 \times 10^3$. Since the original value of Chatterjee *et al.* [2] was fitted to their experimental data, therefore, it is not universal for FXII contact activation. This arbitrary increase of the kinetic rate was motivated to reduce the $\tau_{max}$ value which as shown in Fig 2 lags the experimental data.

$$XII \xrightarrow{\quad} XIIa \tag{3}$$

The third major modification to the Int model is related to FXI activation mechanisms. In the original model, FXI is activated by FXIIa, thrombin and an auto-activation process, respectively:

$$XIIa + XI \leftrightarrow XIIa = XI \rightarrow XIIa + XIa \tag{4}$$

$$XI + IIa \leftrightarrow XI = IIa \rightarrow XIa + IIa \tag{5}$$

$$XIa + XI \leftrightarrow XIa = XI \rightarrow XIa + XIa \tag{6}$$

*In vitro* FXI activation by thrombin is mainly reported with circulated platelets [33] and inhibited by fibrinogen [34]. As TGA experiments were performed on platelet poor plasma (PPP), reaction (5) was deleted from the model. The work of Gailani and Bronze [34] is referenced in the original Int model for reaction (4). Gailani and Bronze [34] stated that the kinetic constants were not determined with precision because it exceeded the achievable factor concentration. Thus, the kinetic constants involved in FXI activation by FXIIa were modified to improve the fit with experimental data ($K_1 = 7 \times 10^8$ M$^{-1}$ s$^{-1}$, $K_{-1} = 200$ s$^{-1}$ $k_{cat} = 0.002$ s$^{-1}$). With the same rationale, the kinetic constant of reaction (6) describing the FXI auto-activation was reduced to an optimal value of $K_1 = 0.7975 \times 10^6$ M$^{-1}$ s$^{-1}$ 4-fold lower than the value reported by Kramoroff *et al.* [33].

Additional modifications were done regarding the cofactor function of FVIIIa. *In silico* thrombin generation parameters showed a larger pro-coagulant profile compared to *in vitro* data, especially for concentrations of FVIII under 1% (see Figs 2 and 4), suggesting that the numerical model overestimated the enzymatic activity of FIXa controlled by reactions (7) and

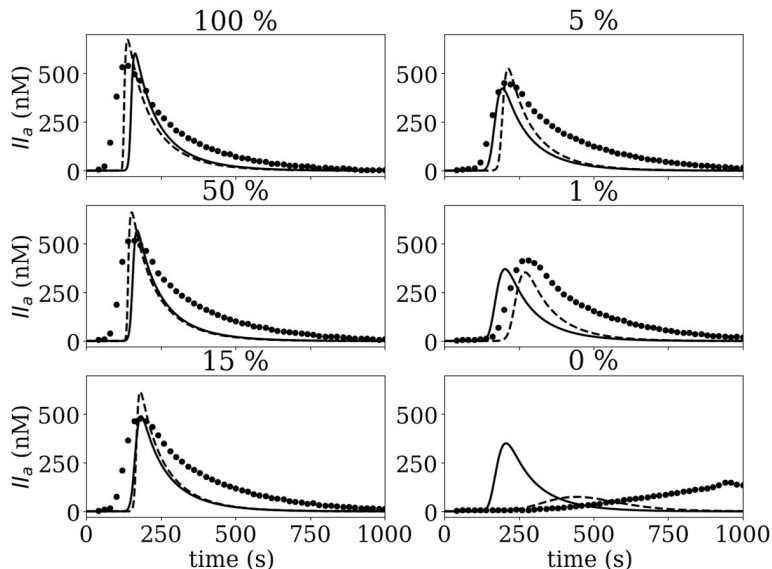

**Fig 4. Thrombin generation (TG) for FVIII concentration range after contact activation the solid line - is the *in silico* data obtained by Int original; the dashed line -- is obtained by Int modified, and • is the experimental data.**

(8).

$$IXa = VIIIa + X \leftrightarrow IXa = VIIIa = X \rightarrow IXa = VIIIa + Xa \tag{7}$$

$$IXa + X \leftrightarrow IXa = X \rightarrow IXa + Xa \tag{8}$$

The kinetic constants of reaction (7) and (8) were modified to decrease the activation of the FX by the unbound FIXa and the intrinsic tenase complex FIXa-FVIIIa. The kinetic constants were obtained from Kogan *et al.* [35] who characterized this reaction focusing on the contact system. The aforementioned modifications improved the numerical model predictions of ETP, $IIa_{max}$, and $\tau_{max}$ for the three ranges of coagulation factors concentrations as observed in Fig 3. The only cases in which modifications reduced the accuracy of the numerical model were the ETP and $IIa_{max}$ values for FVIII. To understand better the underlying behavior in the FVIII case, thrombin production is presented in Fig 4. Fig 4 shows the thrombin production curves for the TGA experiment, Int original, and Int modified simulations. A reduction of the pro-coagulant profile can be observed in the modified data, especially for the 1 and 0% cases. In addition, $\tau_{max}$ has been increased following the trend observed experimentally. The thrombin formation trend for the modified model appears to have a better agreement with experimental data suggesting that the discrepancy observed in ETP and $IIa_{max}$ values of Fig 3 are not indicative of the actual thrombin formation dynamics at low concentrations.

### 3.2 Ext model for the TF pathway

Fig 5 compares the original Ext model thrombin formation to experimental TGA considering physiological plasma. The simulation results show a delayed and reduced thrombin formation compared to the experimental data. This is in line with other numerical studies [21] which reported a thrombin peak formation after 1600 seconds for a physiological plasma sample. In contrast, our experimental TGA thrombin formation peaks around 550 seconds.

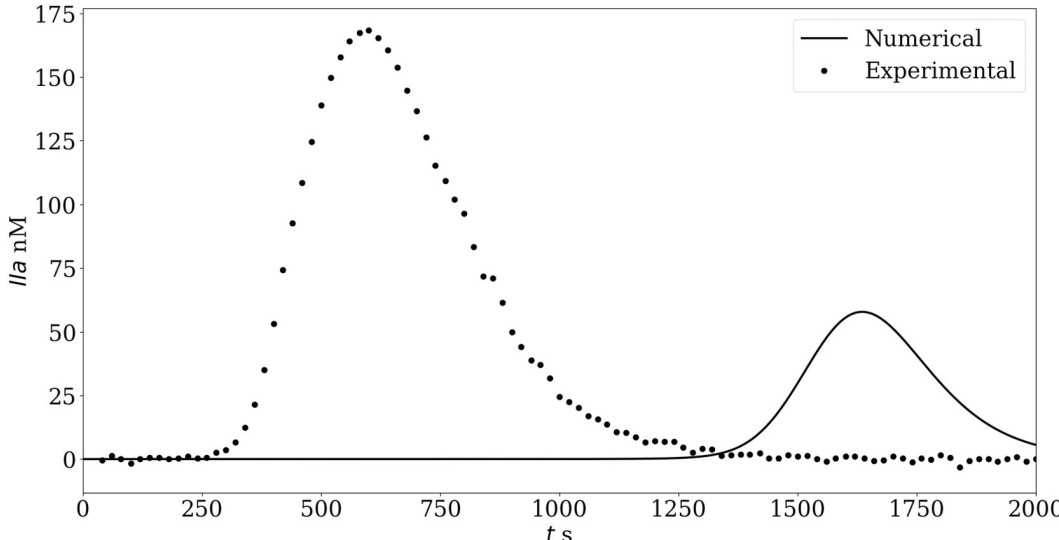

**Fig 5. Thrombin formation in a physiological plasma sample.** The original Ext model shows a low and delayed thrombin generation.

A possible explanation for this observation is that the FVIIa-TF complex formation is independent of calcium concentration and may occur *in vitro* during the TGA pre-incubation phase [36]. In that case, at the beginning of the experimental TGA, the calcium can immediately lead to the activation of FX through the FVIIa-TF complex. To verify this hypothesis, the FVIIa-TF complex concentration was changed *in silico*, modifying the initial concentrations of FVIIa-TF and TF from 0 pM/1 pM to 1 pM/0 pM in both in-silico models. The former values correspond to the nominal value (see Table 1) while the latter correspond to the case where all the TF available has been consumed to form the FVIIa-TF complex before the injection of Calcium at the start of the experiment.

Thrombin formation curves for the original and modified Ext models, as well as experimental TGA data, are displayed in Fig 6 for a range of FVIII concentrations. Remarkable agreement between the numerical and experimental data is observed when the initial value of the FVIIa-TF complex was set to 1 pM.

Fig 7 shows the relative values of ETP, $IIa_{max}$, and $\tau_{max}$ for the original model setting the concentration of FVIIa-TF complex to 1pM, the modified Ext model and experimental data. Note that the agreement between the original model with FVIIa-TF = 1 pM and the experimental values is not good for very small FVIII concentrations ($\leq$ 1%). This observation could be explained by an overestimation of FVIII activity in the Ext model. To explore this hypothesis and improve the activity of the uncomplexed FIXa, the kinetic constants of Eq (8) were modified. An arbitrarily 60 fold increase of the catalytic constant (Kcat = $4.8 \times 10^{-2}$ s$^{-1}$ instead of Kcat = $8.0 \times 10^{-4}$ s$^{-1}$ from [3]). The modified Ext model improved the results remarkably for ETP, $IIa_{max}$ and $\tau_{max}$ variables for the 5, 1, and 0% concentrations. Table 2 summarizes the problems identified to the Int and Ext models and the changes that were introduced.

## 4 Discussion

Numerical models of the coagulation cascade are capable of capturing complex thrombin formation dynamics under physiological conditions. However, more work is needed to improve

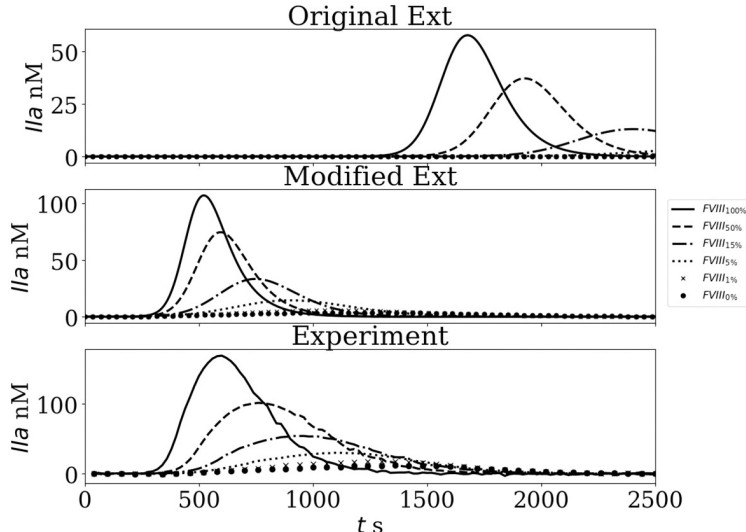

**Fig 6. Thrombin generation (TG) for FVIII concentration range after TF activation A:** *In silico* **TG obtained by the original Ext; B: Obtained by modified Ext; C: Experimental curves.**

their reproducibility and their applicability in different clinical scenarios, as highlighted in the recent review by Chung et al. [38]. In the current work, two well known models of the coagulation cascade were compared with TGA experimental results and proper modifications are suggested to improve their accuracy. The modifications of the Int model for the contact phase are related to excluding reactions that produce spurious delayed thrombin formation due to FVIIa

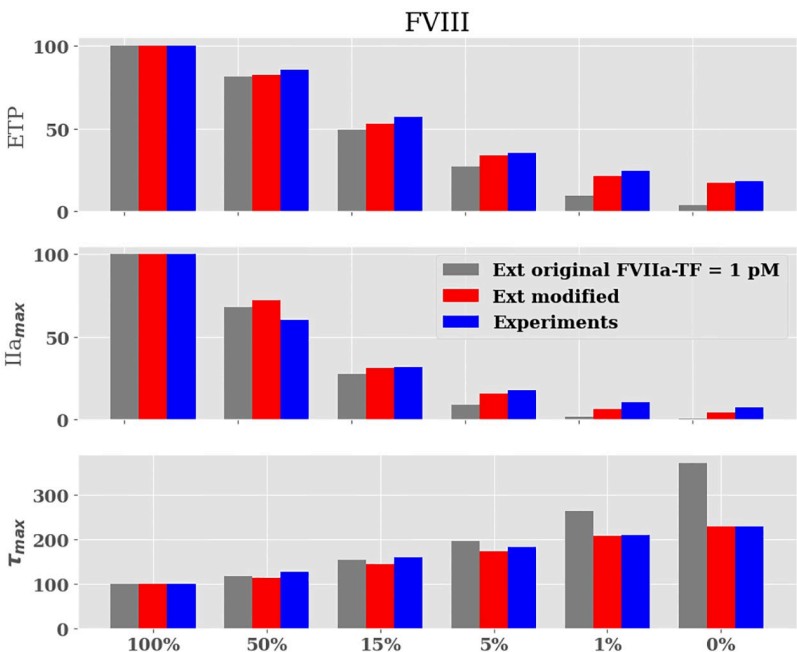

**Fig 7. Evolution of the endogenous thrombin potential (ETP).** GRAY BAR *In silico* ETP obtained by the original Ext model with the initial concentration FVIIa-TF complex = 1pM; RED BAR obtained by modified Ext model; BLUE BAR experimental results.

**Table 2. Modifications performed for Int and Ext models.**

| Model | Problems | Changes made |
|---|---|---|
| Chatterjee *et al.* [2] | • Thrombin generation despite the absence of FXII and FXI<br>• *In-silico* $\tau_{max}$ longer than experimental data<br>• Important thrombin generation decrease when the different FXI rates studied<br>• Excessive thrombin generation for low FVIII concentrations | • reactions (2) and (1) were deleted:<br>VIIa+IX↔VIIa = IX→VIIa+IXa<br>VIIa+X↔VIIa = X→VIIa+Xa<br>• Activation of FXII by reagent<br>$XII \rightarrow XIIa$ ($k_1 = 5 \times 10^{-4}$ to $k_1 = 5 \times 10^{-3}$)<br>• reaction (5) was deleted and kinetic constants of reactions (4) and (6) were modified<br>XI+IIa↔XI = IIa→XIa+IIa (deleted)<br>XIIa+XI↔XIIa = XI→XIIa+XIa; ($K_1 = 7 \times 10^8 M^{-1}s^{-1}$, $K_{-1} = 200 s^{-1} k_{cat} = 0.002\ s^{-1}$)<br>XIa+XI↔XIa = XI→XIa+XIa; ($K_1 = 0.7975 \times 10^6\ M^{-1}\ s^{-1}$)<br>• kinetic constants for reactions (7) and (8) reactions were modified following [35, 37]<br>IXa = VIIIa + X→IXa = VIIIa+Xa; ($K_M = 0.19 \times 10^{-6}$ M, $k_{cat} = 29\ s^{-1}$)<br>IXa+X→IXa+Xa; ($K_M = 2 \times 10^{-6}$ M, $k_{cat} = 0.000667\ s^{-1}$) |
| Butenas *et al.* [3] | • Deficient thrombin generation for low FVIII rate ($< 5\%$)<br>• Delayed thrombin formation due to preincubation of TF-FVIIa | • Kcat = $4,8 \times 10^{-2}\ s^{-1}$ for reaction:<br>IXa + X → Xa + IXa<br>• Initial value of the FVIIa-TF complex was set to 1 pM |

activation in the absence of TF. In addition, FXI activation by thrombin was not considered since platelet poor plasma was used in our study. Then, kinetic constants involved in FXI auto-activation and activation by XIIa were fitted to improve the comparison with TGA results. Another modification regarding excessive procoagulant activity by FVIIIa is proposed to reduce the activation of FX by unbounded FIXa and IXa = VIIIa. As pointed outby Kramoroff et al. [33] and Gailani & Broze [34] the values of the kinetic rates are not precise and thus are susceptible to change. COmpared to the existing Ext model, the main modification consisted in introducing 1 pM initial concentration of FVIIa-TF, which is motivated by the formation of complex FVIIa-TF during the incubation phase of the TGA experiment. This adjustment allowed to shorten the delay of thrombin formation. Lakshmanan *et al.* [39] recently described the same problem of delayed thrombin formation in simulations when compared to experimental data. In their study, the issue was overcome by proposing a feedback activation of FXI and a calibration step of kinetic constants. The delayed thrombin formation was also observed in the study of Pisaryuk et al. [40] in which the original Hockin's model is used to develop a numerical tool to assess individual pharmacokinetic profiles of anticoagulant therapy. Their work showed that simulations differ from the experimental data in terms of lag time and amplitude of the thrombin peak. The authors assumed that these discrepancies were minimal, as the differences were smaller than 15%.

In the current work, the modification on the Int and Ext models improved the comparison for ETP, $IIa_{max}$ and $\tau_{max}$ overall. In the 100% 'nominal' case, the predictions of the original model are fairly good. However, under pathological conditions of FXII, FXI and FVIII deficiencies, the modifications proposed substantially improved the comparison with the experiments and prevented artificial thrombin formation for the 0% concentration cases. Nevertheless, in the case of FVIII ($< 1\%$) triggered via the Intrinsic pathway, both the original and modified Int models fail to reproduce the ETP, $IIa_{max}$ and $\tau_{max}$. Therefore, improvements to the model should be further considered when looking at patients with severe hemophilia conditions.

The role of factor XI in the coagulation cascade is not yet fully understood. Initially set aside in favor of FVIII and FIX, FXI and the contact phase are increasingly taken into consideration following the work of Renné *et al.* in FXII-deficient mice [41] and the search for anti-

XI and anti-XII anticoagulants [42–44], which would be less hemorrhagic than those currently available. In the coagulation models, FXI is not present in the model of Hockin *et al.* The addition of factor XI, and associated reactions as done by Chatterjee *et al.* [2] induces excessive thrombin generation. Lakshmanan *et al.* [39] made the same observation and presented results closer to the assays for low concentrations of tissue factor (0.125 *μ*M and 0.25 *μ*M) by including a feedback activation mechanism of FXI. Nevertheless, the role of FXII is not included in their model, and thus cannot be used in bio-material applications. Chen and Diamond [45] developed a reduced order model to evaluate the role of FXIa and fibrinogen in thrombin and fibrin formation in an open system considering flow transport (both diffusion and convection). They highlighted the thrombin-feedback activation of FXIa in thrombin formation pointing out again an important role of FXI in the coagulation cascade. As shown in the current work, the model of Hockin *et al.* responds well to FVIII deficiency. However, due to the lack of modeling regarding FXI activity, the results can be misleading, as pointed out by previous studies and the current results.

The proposed modifications to the Int and Ext models are not universal (as shown in reaction 8 with different values for Int and Ext models) but are well suited for the TGA experimental conditions studied. Thus, they can be used as a valuable tool to explore any scenario that might be costly experimentally. Since the coagulation model still has limitations linked to modeling assumptions, one should be careful when extrapolating the models to open system applications such as thrombosis simulations of medical devices or organ scale studies. Considering the complexity of kinetic schemes of the coagulation cascade, the authors recommend close collaboration between hematologists and modelers when applying coagulation models. Continuous validation of coagulation models in complex situations such as pathological cases is needed in order to broaden their application scope. This remains a necessary effort towards robust and reliable coagulation models that could be used to study coagulation disorders and in drug or bio-medical devices development.

## Supporting information

**S1 Text. Coagulation reaction schemes used in the original and modified models.** (PDF)

## Acknowledgments

The authors would like to thank Prof. Jean-François Schved for his insightful comments.

## Author Contributions

**Conceptualization:** Alexandre Ranc, Muriel Giansily-Blaizot, Franck Nicoud, Rodrigo Méndez Rojano.

**Data curation:** Alexandre Ranc, Salome Bru, Rodrigo Méndez Rojano.

**Formal analysis:** Alexandre Ranc, Simon Mendez, Muriel Giansily-Blaizot, Rodrigo Méndez Rojano.

**Funding acquisition:** Franck Nicoud, Rodrigo Méndez Rojano.

**Investigation:** Alexandre Ranc, Franck Nicoud.

**Methodology:** Alexandre Ranc, Franck Nicoud.

**Project administration:** Simon Mendez.

**Resources:** Simon Mendez, Muriel Giansily-Blaizot, Franck Nicoud.

**Software:** Alexandre Ranc, Simon Mendez, Franck Nicoud, Rodrigo Méndez Rojano.

**Supervision:** Simon Mendez, Muriel Giansily-Blaizot, Franck Nicoud, Rodrigo Méndez Rojano.

**Validation:** Alexandre Ranc, Salome Bru, Rodrigo Méndez Rojano.

**Visualization:** Salome Bru, Rodrigo Méndez Rojano.

**Writing – original draft:** Alexandre Ranc, Rodrigo Méndez Rojano.

**Writing – review & editing:** Simon Mendez, Muriel Giansily-Blaizot, Franck Nicoud, Rodrigo Méndez Rojano.

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
