## [Decision Letter · Decision Letter 0]

21 Mar 2023

PONE-D-23-04356Critical evaluation of kinetic schemes for coagulationPLOS ONE

Dear Dr. Méndez Rojano,

Thank you for submitting your manuscript to PLOS ONE. After careful consideration, we feel that it has merit but does not fully meet PLOS ONE’s publication criteria as it currently stands. Therefore, we invite you to submit a revised version of the manuscript that addresses the points raised during the review process.

We look forward to receiving your revised manuscript.

Kind regards,

Akbar Dorgalaleh

Academic Editor

PLOS ONE

Journal Requirements:

2. Please expand the acronym “CONACYT and  ANR ” (as indicated in your financial disclosure) so that it states the name of your funders in full. This information should be included in your cover letter; we will change the online submission form on your behalf.

Reviewers' comments:

Reviewer's Responses to Questions

**Comments to the Author**

1. Is the manuscript technically sound, and do the data support the conclusions?

Reviewer #1: Yes

Reviewer #2: Yes

Reviewer #3: Yes

2. Has the statistical analysis been performed appropriately and rigorously? 

Reviewer #1: Yes

Reviewer #2: No

Reviewer #3: N/A

3. Have the authors made all data underlying the findings in their manuscript fully available?

Reviewer #1: Yes

Reviewer #2: Yes

Reviewer #3: Yes

4. Is the manuscript presented in an intelligible fashion and written in standard English?

Reviewer #1: No

Reviewer #2: Yes

Reviewer #3: Yes

5. Review Comments to the Author

Reviewer #1: Dear Editor of the PLOS ONE Journal

The manuscript entitled “Critical evaluation of kinetic schemes for coagulation” is an interesting paper because the authors, in addition to presenting the problem in this area, have well illustrated its solution, both in the internal and external coagulation pathways regarding the thrombin generation and thus it certainly deserves publication into the journal, But to meet the increasingly high-quality standard of the Journal some minor revision is needed according to the following points. Furthermore, the authors should consider some grammatical errors correction.

Comments

1. Using a phrase like "hemophilia A patient" stigmatizes the patient and should be used as " patient with hemophilia A" to maintain the patient's respect.

2. “inhouse” OR “in-house”. Please use a single form of the word in the text.

3. The sentence “%, and (iv) the lag time (τlag) corresponding to the time required for the generation of 10 nM of thrombin” is not clear. Please rewrite the sentence and specify the role of "%" and determine the expanded form of (iv).

4. The word abbreviations should be expanded in all figures and tables.

5. In the vertical axis of Figure 1, unit (M) is used for unitization. But at the same time, nM is also displayed in this axis, the reason for which is not clear. In addition, the number 10-7 is also mentioned in its upper part, which carries a similar situation. Authors should clarify this.

6. The reason for using Ext and Ent is explained in lines 130 and 131, while lines 65 and 67 refer to these words. Please move this sentence to its original place.

7. in this sentence "Two coagulation kinetic models were used in the current study. For the sake of simplicity, the models of Chatterjee et al. [2] and Butenas et al. [3] are named Int and Ext, respectively, making reference to the intrinsic and extrinsic pathway." The word "pathway" should be used in plural.

8. In line 156 “The rationale for each specific modification is discussed in Section” (And the same in table 1). Section not defined

9. In lines 160 and 178, after IIamax, insert “and” and in line 356, “thombin” should be corrected as “thrombin”

10. The discussion section should provide more details using the results of the study compared to other research

11. Some of references are not arranged according to journal guidelines. They also are not arranged in the same manner. The author should review and correct all of them.

Reviewer #2: The authors compared the intrinsic and extrinsic pathways of coagulation with thrombin generation assays considering realistic pathological conditions. In my opinion, this manuscript can be accepted after applying the requested amendments.

Comment 1:

-The abstract is somewhat unintelligible; it is hard to understand what are the aim, methods, results and conclusion of the study.

Comment 2:

-In general, the article is written complex and should be written in a simpler and more understandable way. For example,

page 1, lines 11, 12, 13

Numerical representations of the coagulation cascade aim to mimic the thrombin generation process which is the result of the balance between prothrombin conversion and thrombin inactivation [9], thrombin being the key enzyme of the blood clotting cascade.

Comment 3:

-Using a combination of different words for the same concept will be confusing; it is suggested to use the same words. For example,

Numerical representations

numerical modeling

Numerical data

numerical cases

numerical thrombin production

…

Comment 4:

-What does (. %,) mean in the following line? If it is wrong, correct it.

page 4, line 125

… (iii) the time to peak (τmax) which is the time required to reach IIamax [20]. %, and (iv) the…

Comment 5:

page 13, line 325

As pointed out in [25, 26] the values of…

-I suggest that the names of the authors be used.

Comment 6:

-The first paragraph of the discussion section is suitable for the introduction. It is better to merge this part with the introduction. This change will lead to a better understanding of the purpose of the study.

Reviewer #3: Despite the fact that numerical modeling of the coagulation cascade has a long history at the moment there are gaps in this area of research. The results of numerical modeling should be the study of the pharmacokinetics of new drugs and the study of the dynamics of thrombosis in pathological conditions. Well-validated quantitative models of the coagulation cascade are expected to complement traditional laboratory as predictive tools in clinical practice, enabling physicians to estimate disease risk or simulate therapeutic outcomes in individual patients. In this paper, the authors evaluated two well-established mathematical models of the coagulation cascade for the contact pathway and for the TF path in conditions of clotting factor VIII, XI or XII deficiencies. The authors describe the modifications made for the Int and Ext models. The proposed modifications to the Int and Ext models can be used as a valuable tool to explore any scenario in a less expensive way compared to the experimental path. Given the complexity kinetic schemes of the coagulation cascade, the authors call for close cooperation between hematologists and modelers in the application of coagulation models.

The article has a traditional structure. A sufficient number of tables and figures make it easier to understand the work done. The list of references is represented by a large number of publications from the period from 1990 to 2021.

6. PLOS authors have the option to publish the peer review history of their article (what does this mean?). If published, this will include your full peer review and any attached files.

Reviewer #1: No

Reviewer #2: No

Reviewer #3: No

---

## [Author Response · Author response to Decision Letter 0]

12 Jun 2023

We have carefully read your diligent comments about our manuscript and we have done our best to address them. We have added a Rebuttal Letter in the submission in which we detail a point-by-point answer to the reviewers’ comments (see page 43). A marked up manuscript with changes in blue is included along with an unmarked manuscript as requested by the journal.

---

## [Decision Letter · Decision Letter 1]

11 Aug 2023

Critical evaluation of kinetic schemes for coagulation

PONE-D-23-04356R1

Dear Dr.Rodrigo,

We’re pleased to inform you that your manuscript has been judged scientifically suitable for publication and will be formally accepted for publication once it meets all outstanding technical requirements.

Kind regards,

Gausal Azam Khan, Ph.D;CSci,FRSB

Academic Editor

PLOS ONE

Additional Editor Comments (optional):

Reviewers' comments:

Reviewer's Responses to Questions

**Comments to the Author**

1. If the authors have adequately addressed your comments raised in a previous round of review and you feel that this manuscript is now acceptable for publication, you may indicate that here to bypass the “Comments to the Author” section, enter your conflict of interest statement in the “Confidential to Editor” section, and submit your "Accept" recommendation.

Reviewer #1: All comments have been addressed

Reviewer #2: All comments have been addressed

2. Is the manuscript technically sound, and do the data support the conclusions?

Reviewer #1: Yes

Reviewer #2: Yes

3. Has the statistical analysis been performed appropriately and rigorously? 

Reviewer #1: N/A

Reviewer #2: I Don't Know

4. Have the authors made all data underlying the findings in their manuscript fully available?

Reviewer #1: Yes

Reviewer #2: Yes

5. Is the manuscript presented in an intelligible fashion and written in standard English?

Reviewer #1: Yes

Reviewer #2: Yes

6. Review Comments to the Author

Reviewer #1: (No Response)

Reviewer #2: (No Response)

7. PLOS authors have the option to publish the peer review history of their article (what does this mean?). If published, this will include your full peer review and any attached files.

Reviewer #1: No

Reviewer #2: No

---

## [Editor Report · Acceptance letter]

17 Aug 2023

PONE-D-23-04356R1 

Critical evaluation of kinetic schemes for coagulation 

Dear Dr. Méndez Rojano:

I'm pleased to inform you that your manuscript has been deemed suitable for publication in PLOS ONE. Congratulations! Your manuscript is now with our production department. 

Kind regards, 

on behalf of

Prof. Gausal Azam Khan 

Academic Editor

PLOS ONE